# Sources of Resilience in Frontline Health Professionals during COVID-19

**DOI:** 10.3390/healthcare9121699

**Published:** 2021-12-08

**Authors:** Lydia Brown, Simon Haines, Hermioni L. Amonoo, Cathy Jones, Jeffrey Woods, Jeff C. Huffman, Meg E. Morris

**Affiliations:** 1Victorian Rehabilitation Centre & Healthscope, Glen Waverley, VIC 3150, Australia; l.brown@latrobe.edu.au (L.B.); jeffrey.woods@healthscope.com.au (J.W.); 2Academic Research Collaborative in Health, La Trobe University, Melbourne, VIC 3086, Australia; s.haines@latrobe.edu.au (S.H.); cathyjoneshealth@gmail.com (C.J.); 3Melbourne School of Psychological Sciences, University of Melbourne, Melbourne, VIC 3010, Australia; 4Harvard Medical School, Harvard University, Boston, MA 02115, USA; Hermioni_Amonoo@DFCI.HARVARD.EDU (H.L.A.); JHUFFMAN@mgh.harvard.edu (J.C.H.); 5Department of Psychiatry, Brigham and Women’s Hospital, Boston, MA 02115, USA; 6Department of Psychosocial Oncology and Palliative Care, Dana-Faber Cancer Institute, Boston, MA 02215, USA; 7Department of Psychiatry, Massachusetts General Hospital, Boston, MA 02114, USA

**Keywords:** COVID-19, health professionals, workforce, behaviour

## Abstract

Background: While the challenges for psychological well-being for Australian healthcare workers have been documented, there has been a dearth of qualitative research on the sources of resilience that sustained workers during the COVID-19 pandemic. This study identified sources of resilience that clinicians used to cope with frontline challenges during the COVID-19 pandemic. Methods: Semi-structured interviews were conducted with 20 frontline health professionals, across five Australian hospitals, between October 2020 and April 2021. The interviews were recorded and transcribed, and the results were analysed using thematic analysis based on a phenomenological approach. Results: Three sources of resilience were identified by respondents: personal, relational, and organisational. A positive mindset, sense of purpose, and self-care behaviours emerged as key sources of personal resilience. Teamwork, altruism, and social support from family and friends contributed to relational resilience. Leadership, effective communication, and effective implementation of COVID-19 policies were associated with resilience at the organisational level. Frontline healthcare workers also voiced the need for the implementation of further strategies to support personal resilience whilst nurturing resilience within clinical teams and across entire healthcare organisations. Conclusions: Trust in healthcare systems, organisation leaders, colleagues, and personal support teams was an overarching theme supporting resilience.

## 1. Introduction

Frontline healthcare workers in Australia, and internationally, have faced unprecedented demands during the COVID-19 pandemic that have tested their resilience. Resilience refers to responding to a potentially highly disruptive event adaptively so that psychological and physical functioning is preserved and personal growth occurs [1]. Resilience enables healthcare workers to ameliorate stress, reduce their risk of burnout, and maintain wellbeing during challenging times [2,3].

There has been a rapid accumulation of questionnaire-based quantitative research showing that healthcare worker resilience during COVID-19 has been shaped by a range of demographic, psychological, and work-related factors [4,5,6,7]. The level of virus exposure, country-wide infection rates, availability of personal protective equipment, vaccinations, excessive exposure to negative media reports, and cultural differences explain some of the individual differences in stress, anxiety, and coping [5,6,8,9,10,11,12]. Relative to significant quantitative research on clinician resilience and coping, comparatively little is known about the lived experience of resilience experienced by frontline health professionals during the pandemic assessed using qualitative research methodology.

To date, two qualitative studies have specifically aimed to investigate frontline clinician experiences of resilience during the COVID-19 pandemic [9,10]. These studies were based on interviews with Indian [9] and Pakistani [10] frontline healthcare clinicians, respectively. Banerjee and colleagues (2021) interviewed 172 frontline health professionals working in India and showed that a large proportion had a fear of becoming infected, as well as feelings of anxiety, distress, guilt, stigma, and isolation [9]. Banerjee et al. (2021) highlighted the importance of fostering positive emotions and mental wellbeing during the pandemic, by implementing flexible workplace policies and by ensuing physical protection from the virus. Results from this study also found that social networks, peer support, and a focus on self-care helped to foster resilience [9]. Likewise, Munawar and Choudhry [10] used thematic analysis in a qualitative study to understand stress and coping in 15 COVID-19 frontline emergency healthcare workers in Pakistan. Coping strategies included limiting exposure to negative media, drawing upon religious beliefs, and taking strength from their motivation to serve their patients and country. To our knowledge, aside from these two studies from South Asia, no other qualitative studies have specifically focused on sources of resilience that clinicians have utilised during the pandemic; however, coping and growth have emerged as themes in more general studies of clinician experiences during COVID-19 times [11,12,13,14,15]. Thus, there is a need to better understand sources of strength that clinicians draw on to manage the challenges of the pandemic in different cultural settings, including Australia.

Despite limited interview-based research on clinician resilience, COVID-19 policy recommendations have been made to protect clinician well-being during the pandemic [16]. Furthermore, interventions have been developed to promote healthcare clinician well-being at this time [17,18]. To compliment this work, there is a need to identify sources of strength and resilience that clinicians can draw upon during the pandemic [2,19], and this paper aims to achieve this goal by asking clinicians about their direct experiences of resilience using qualitative methodology.

Given the lack of qualitative research specifically focused on clinician sources of resilience during the pandemic globally, and also in Australia, the aim of this investigation was to explore sources of resilience among Australian frontline healthcare workers. Sources of resilience included factors that facilitated physical and emotional functioning and meaningful personal growth during the COVID-19 pandemic. We conducted individual in-depth interviews with medical practitioners, nurses, and allied health practitioners to elicit their views about sources of resilience that optimised functioning and well-being during the COVID-19 crisis.

## 2. Materials and Methods

### 2.1. Design, Setting, and Participants

The use of thematic analysis, an inductive, phenomenological approach [20,21], enabled key themes to be captured from the transcripts of semi-structured, individual interviews of approximately 35–45 min duration, conducted via videoconference. Participants were recruited via email, or in person at staff meetings at five hospital sites between October 2020 and March 2021, using purposeful sampling and were required to be frontline healthcare professionals (nurses, medical practitioners, or allied health clinicians). To be recruited, they needed to be in clinical roles in hospitals with COVID-19 patients and to be fluent in the English language. We define frontline healthcare professionals as those working in hospitals with COVID-19-positive patients. Twenty-one participants were invited for an interview, out of whom one subsequently declined for personal reasons.

### 2.2. Procedure

Participants provided informed consent electronically via REDCap software, completed a demographic questionnaire and self-report measures of stress and resilience, and were then scheduled for an interview via videoconference or telephone. Interviews followed a pre-prepared interview guide and were conducted by the first author, a registered clinical psychologist (LB). Interviews were audio-recorded and transcribed verbatim. Participation was voluntary and not incentivised.

We adhered to the consolidated criteria for reporting qualitative research (COREQ) guidelines and received ethics approval from La Trobe University’s Human Research Ethics Committee (HEC20385). It was prospectively registered on the Australia and New Zealand Clinical Trials Registry (ACTRN12620000921987p).

### 2.3. Data Collection

Frontline healthcare workers were recruited between October 2020 and March 2021. Prospective participants were informed about the study via email or in person at staff meetings at eligible hospital sites. Those who were interested in participating provided informed consent electronically via the REDCap system and completed a short demographic questionnaire as well as tests of stress and resilience. Participants were then contacted via email to arrange an interview via videoconference or telephone. Semi-structured interviews followed a pre-prepared interview guide (see Appendix A) and were conducted by a registered clinical psychologist and member of the research team (LB). Interviews were audio-recorded and then transcribed verbatim.

### 2.4. Data Management and Analysis

The demographic data and self-report measures of stress and resilience were captured using REDCap software. The Perceived Stress Scale (PSS-10), a widely used, valid and reliable measure [22], was used to index stress and showed excellent internal reliability, α = 0.89. To measure resilience, the Connor–Davidson Resilience Scale (CD-RISC-10), a valid and reliable measure of resilience in both healthy and clinical populations [23], was used, and also showed excellent internal consistency, α = 0.88.

Interview transcriptions were independently coded using NVivo, and thematic analysis was used to analyse the coded interview data, based on the methods described by Braun and Clarke [20], Castleberry and Nolen [21], and Vaismoradi, Turunen, and Bondas [24]. Coders held weekly meetings to ensure inter-rater reliability and discuss emerging themes. When consensus could not be achieved, a third expert (MEM) was consulted, until agreement occurred.

We minimised the risk of bias and ensured study rigor by (i) documenting our methodology prior to data collection; (ii) ensuring that at least two healthcare professionals from our research team conducted the data analysis; (iii) putting in place a protocol to verify accurate recording and transcription of the interview recordings; (iv) checking that recruitment was confined to health professionals who had worked on the COVID-19 frontlines; (v) collecting data until saturation was reached; (vi) utilising participant quotes to generate themes [25,26].

## 3. Results

### 3.1. Participant Characteristics

Participating frontline healthcare professionals included eight nurses, five physicians, and seven allied health clinicians. The allied health clinicians were physiotherapists (3), occupational therapists (2), and psychologists (2). Fifteen (75%) had worked or were currently working with COVID-19-positive patients. They were highly experienced, with 17 (85%) working for at least 10 years in their profession. Most (*n* = 14, 70%) worked full-time. Participants reported low to moderate perceived stress (M = 12.7; SD = 6.2) relative to a large sample of frontline medical personnel working in the COVID-19 setting [27]. There were similar levels of resilience (M = 32.1; SD = 5.3) to a recent study using the CD-RISC-10 with physicians working in a critical care ward [28]. Participant characteristics are summarised in Table 1.

### 3.2. Qualitative Results

Participants described sources of resilience on three nested levels: personal, relational, and organisational, as summarised in Figure 1.

#### 3.2.1. Theme 1: Personal Resilience

##### Mindset

Frontline healthcare workers advised that their mindset and attitudes helped them to adjust to the challenging work conditions of the pandemic: “I think keeping a positive mindset certainly helped. It could have been a really difficult time if, I guess, we were thinking negatively all the time.” Specifically, adopting a positive outlook, capacity to flexibly adapt to rapidly changing circumstances, and an action-oriented approach were common psychological factors raised by participants. To think positively, participants emphasised the importance of hope, a light at the end of the tunnel: “I think hope is really important during these times of struggles because it’s easy to think there is no letting up.”

Some indicated that it was essential to flexibly adapt to changing work conditions: “you had to adjust your mindset to what your different priorities were.” The capacity to be flexible did not always entail effortful thinking. Instead, frontline healthcare professionals often prioritised action over thought in order to respond to urgent clinical needs “especially when you’re busy, you don’t have time to think about anything, you just do it.”

##### Finding a Sense of Purpose

The participants voiced that their contribution to the pandemic response was meaningful: “It just gave me a sense of usefulness and purpose for sure”; “it was also this once in a lifetime experience to be part of something bigger than ourselves.” Their contribution generated a sense of accomplishment: “The tears probably come from pride actually because we made it and we’ve done such a good job”, as well as personal and professional growth.

In terms of personal growth, many frontline healthcare professionals described the discovery of hidden resources, uncovering strengths within themselves: “This year, look, has definitely been the most challenging year I’ve had. But I think it’s surprised me at how strong I can be.” Some described the capacity to translate growth derived from pandemic to life outside of work, indicating potential for lasting, positive change: “I have retained more resilience in my mindset with everyday interactions, I think. I haven’t gone—I haven’t spiralled back into self-doubt in the way that I had pre-pandemic.”

Several described professional growth including gaining new skills: “it really centred me. It made me better at my job, my executive job, and it was just a place I could concentrate at the very base level of being a doctor.” Increased sense of trust in one’s own abilities, as well as perceived trust from colleagues, was also noted: “People have started to trust me because I demonstrated that I’m capable, actually I can do this. I’m not the most confident person normally but people count on me a bit more.”

##### Self-Care Behaviours

Almost all of the frontline healthcare professionals who were interviewed (17/20) reported that self-care behaviours and routines or rituals, including exercise and contemplative practices, helped them cope and foster mental balance during the pandemic. Going for walks and other physical activity was often noted as a helpful behaviour, even for those who did not typically exercise: “I’m not a very big exercise person, I don’t work out much, but this time walking really helped me”; “having to just take care of myself physically, I think, helps as well because if I don’t feel like I’ve got the physical energy, I’m not going to have the emotional energy.” Contemplative practices such as meditation, yoga, and prayer were also used by a number of participants to foster mental balance: “I have meditated off and on over the years and that’s something I have reintroduced during this time, just to help me calm down a little bit.”

#### 3.2.2. Theme 2: Relational Resilience

##### Teamwork

Teamwork was a central and recurring theme: “Everyone looked after each other. Everyone had each other’s back.” The participants reflected on the necessity of trusting their colleagues to work together and carefully follow infection control procedures: “For the most part, the team were so close-knit and so onto their PPE... I felt very strong in my PPE, and I felt very strong working as part of a team that were on the COVID ward.” Some indicated that teamwork was in excess of what they were accustomed to outside of the pandemic, and it strengthened over time: “as the time went on, it just changed so dramatically, to the point where it was just such comradery between the nursing staff and everybody who was in that team, and the patients.” They indicated a deep emotional connection to their teammates, related to bonding through trauma: “I think that experience will bond us forever in a way, the team there. You do bond in trauma, and it was trauma for us.” The reported connection often transcended professional boundaries, and some participants found their colleagues became more like family: “So we went out to ward and we experienced a lot together and it just connected us. And we were still very connected every time we see each other. We’re like family now.”

##### Altruism

Many described acts of altruism or transcendence where they or their teammates went above and beyond the call of duty to help others during the pandemic: “when the chips are down, people usually just jump in and do what they can do to help.” Frontline healthcare professionals often provided emotional support to patients in lieu of family. They were forced to manage end-of-life issues with patients in a way that was beyond the regular provision of emotional support. “One resident was passing, and the daughter said to me, “Somebody must sit behind my mom.” … I said, “They come in and they sit.” She says, “I can see by the chairs.” The chair is pulled up so she can see the sides. Because they knew the room so well. And I said to her, “The night nurse that was on,” I said, “I know she’s held your mom’s hand.” I said, “Granted she’s had to hold it through a glove, but she’s held your mom’s hand and your mom knows she’s there.” And to that daughter that just meant everything.”

Medical, nursing and allied health practitioners acknowledged the dual medical and emotional needs of COVID-19-positive patients: “There was medical care that was needed, but it was tender, loving care.” Staff also went above and beyond to support each other. For instance, staff who did not have direct contact with infected patients helped in other ways, such as preparing food: “Staff that couldn’t come—food. You know, hospital staff had food. There was just food. They knew that it had to be individually wrapped and there were boxes of individually wrapped food.”

##### Social Support Outside of Work

Participants drew on social support from family and friends for buoyancy during frontline work of the pandemic. Some found solace in emotionally debriefing: “Having my family, my friends, and my husband to talk to afterwards was really helpful.” Others appreciated family understanding of increased workload: “My family adapted to the fact that I had greater responsibility and greater need to be at work … That did help enormously.”

Notably, some participants acknowledged their family’s support of the risk of infection that they were taking on by contributing to the pandemic response: “Regardless, my mother said, ‘If I die because of you, I said yes that you could help. That’s okay by me’; “I could easily think if my family would have said very strongly that they didn’t want me to do it, I don’t know how I would have taken to that situation.”

#### 3.2.3. Theme 3: Organisational Resilience

The health professionals discussed the importance of hospital-level organisational factors, with a central theme of the need for trust in the design and implementation of policies and procedures, trust in leadership, and effective communication platforms across different levels of management and practice.

##### Design and Implementation of Policies and Procedures

Participants in senior positions emphasised a primary focus on establishing infection control policies from the beginning of the pandemic: “From early on, it was really just putting together the pandemic plan.” Participants also discussed the setup of an incident command centre and work to clarify clear lines of command: “I asked him to be the incident commander because I thought that made sense... It allowed me to be involved but not having to control a lot of the mechanics of it in terms of who’s right, who’s got the pen on that policy.”

Participants also spoke about the importance of adherence to, and trust in, the procedures established by management: “I felt very strong and very protected by policies and procedures that were developed quickly at the beginning of the pandemic.” Trust in and implementation of procedures helped mitigate staff fear of contagion “when I talked to people on the ward, they didn’t seem afraid, because we felt very strong in our PPE, and we felt very strong in the processes.”

##### Leadership

Effective leadership was discussed in detail by participants. The five healthcare leaders spoke about the importance of finding ways to be present for their team, often going above and beyond usual practice to help instill a sense of trust and stability: “They knew I was here night and day, so they felt that I was here, and I was a presence, so that was important. I’m fairly present anyway, but it was important for me to do those long days initially, because they needed to know I was here with them.” Staff seemed to appreciate visual leadership: “Our director of nursing … they were on the forefront just as much as we were, so they led from the front, we led from the front, we had to.” In addition to trust, some leaders advised that they could better understand and improve processes by leading from the front: “It sounds sort of paradoxical but going into those hot zones allowed me not only to help redesign them and sort of tweak a lot of it—but also to create tools for the frontline staff to use effectively.” This contributed to the leader’s own resilience during the pandemic, by fostering teamwork and feelings of pride.

Leaders also spoke about a responsibility to be a positive role model for their team, despite their own insecurities: “Because I saw my team as looking to me for an example, and to lead them through with integrity, compassion, and honesty. Then I just had to do that. There was no other option, except fall apart, and that wouldn’t have served the people who were looking to me as their leader to lead through the pandemic.”

##### Effective Communication

Dovetailing with the theme of effective leadership, communication across different levels of the hospital helped participants to feel supported: “the whole corporate team threw themselves into it as well, so I had all this support, this amazing support and talking to people that I’d only ever seen their names on an email address, talking to people on Zooms.” Communication ensured that people could place their trust in a structured, coordinated pandemic response: “So it was getting structure, it was getting people trusting that structure, so after a while we got into a rhythm and they understood that there was a forum to be heard.”

Participants voiced the necessity of utilising new methods of communication to adjust to rapidly changing circumstances and policy changes during the pandemic: “We did set up a lot of different communication tools, the Zoom, Facebook, and any other way we could try and get the messages out. And that became a really powerful tool, all the way through it.” Communication platforms also helped frontline healthcare professionals seek support and guidance from colleagues: “We did have this WhatsApp group where we talk to each other. If we have any difficulty, we share that difficulty.”

#### 3.2.4. Trust

Trust was a higher-order mechanism that contributed to resilience at each of our nested levels depicted in Figure 1. Personally, participants reported the importance of trusting both their professional training and also their inner mental strength. In turn, trust fuelled a healthy mindset, as well as personal and professional growth. Relationally, frontline healthcare professionals placed trust in their colleagues’ use of PPE, and more broadly, they frequently reported trusting that their teammates ‘had their back’, facilitating teamwork. At the organisational level, trust was a key ingredient relevant to leadership, effective communication, and successful implementation of policies and procedures. Based on study findings, we provide a list of recommendations that healthcare organisations can adopt to foster resilience (Table 2).

## 4. Discussion

This study showed the lived experience of resilience of healthcare workers to be shaped by personal, interpersonal, and work-related factors [29]. Positive coping during the pandemic was associated with having strong social support as well as organisational factors such as effective communication. These findings agree with the scoping review by Rieckert and colleagues [7]. A key finding of our study was that frontline healthcare professionals were able to take active steps themselves to nurture a positive mindset and to reflect on the significance of their work and enact self-care behaviours. As with recent US policy recommendations [16], relational and organisational factors were found to play major roles in empowering frontline healthcare professionals to navigate the challenges of being front-line workers caring for people with the highly contagious COVID-19 virus.

Our findings respond to a call from Holmes and colleagues [2] to identify modifiable sources of resilience and to provide data to advocate for the need to safeguard mental health in frontline healthcare professions [30,31]. Quantitative studies have modelled the determinants of resilience in health professionals [5,6,8,32,33,34]. For instance, a UK and Ireland study of 1305 frontline workers by Sumner and Kinsella [6] found that psychological factors, exposure to COVID-19, and country of residence were significant predictors of resilience [6]. Another UK study of 255 frontline healthcare professionals found people who were older and with extensive professional experience had greater resilience during the COVID-19 pandemic [8]. While these findings help shed light on some of the contributors to resilience, many of the determinants of resilience, such as country of residence, years of experience, and age, are not readily modifiable. By adopting a qualitative research design, we were able to elucidate lived experience of resilience during the pandemic.

Our findings also compliment the work by Shanafelt and colleagues [13], where US healthcare professionals were interviewed about their sources of anxiety during the pandemic. By adopting a positive psychology framework [19], we add a complementary perspective that sheds light on experiences of strength and resilience that can be used to rise to some of the challenges of pandemic work [13]. We found that frontline healthcare professionals can draw strength within themselves. Additionally, resilience can be fostered relationally at the level of teams. It can be supported at the organisational level via effective policies and procedures, communication, and leadership.

Based on these findings and global research [2,7,16,35], we recommend that organisations take action to promote resilience at the level of individual clinicians, clinical teams, and across the organisation as a whole, instilling trust at each level (Table 2). By adopting multiple and complimenting strategies, staff can be supported at times when their skills are acutely needed to protect the health and well-being of society as a whole (Table 2). As part of a national strategy to protect frontline healthcare professional well-being during COVID-19, the New England Journal of Medicine recently recommended that healthcare organisations appoint ‘Chief Wellness Officers’ within their COVID-19 command structure [16]. A chief wellness officer could, for example, oversee and implement resilience-building strategies (Table 2). This could help safeguard clinician mental health, which is especially important given clinicians have been identified as a group vulnerable to mental health symptoms during the pandemic [36].

There were some limitations of this qualitative study. The data were collected across five Australian hospitals where frontline healthcare professionals had the opportunity to opt out of the frontline COVID-19 response. Thus, while all participants were frontline professionals in that they worked in a COVID-19-positive hospital, a minority of participants (25%) did not directly work with COVID-19-positive patients. For this reason, it is uncertain whether the findings generalise to other countries or settings where frontline work is compulsory.

A second important limitation of findings from this study is that data were collected in 2019, during the first phase of the pandemic when the lack of personal protective equipment, lack of confidence in safety measures and fear of infections were predominant. Since this time, the picture of psychosocial factors affecting clinician wellbeing has changed significantly, and new themes such as isolation, compassion fatigue, and monotony have become important [37]. For instance, a longitudinal quantitative study by Magnavita and colleagues has tracked changes in frontline clinician experiences over time between May 2020 and May 2021 [37,38]. The authors found that doctors reported increasing time restraints for self-nurturing activities such as physical activity and meditation over the one-year period. Thus, it is important to consider barriers to resilience and to acknowledge that some of these barriers may have increased with time over the course of the pandemic. Consideration of modifiable resilience factors, such as emotional intelligence [39], is warranted to help clinicians navigate the evolving challenges of the pandemic.

## 5. Conclusions

To conclude, frontline healthcare workers perceived a need for organisational, relational, and personal support to build and maintain resilience during the COVID-19 pandemic. They identified a need for further strategies to support personal resilience whilst nurturing resilience within clinical teams and across organisational structures.

## Figures and Tables

**Figure 1 healthcare-09-01699-f001:**
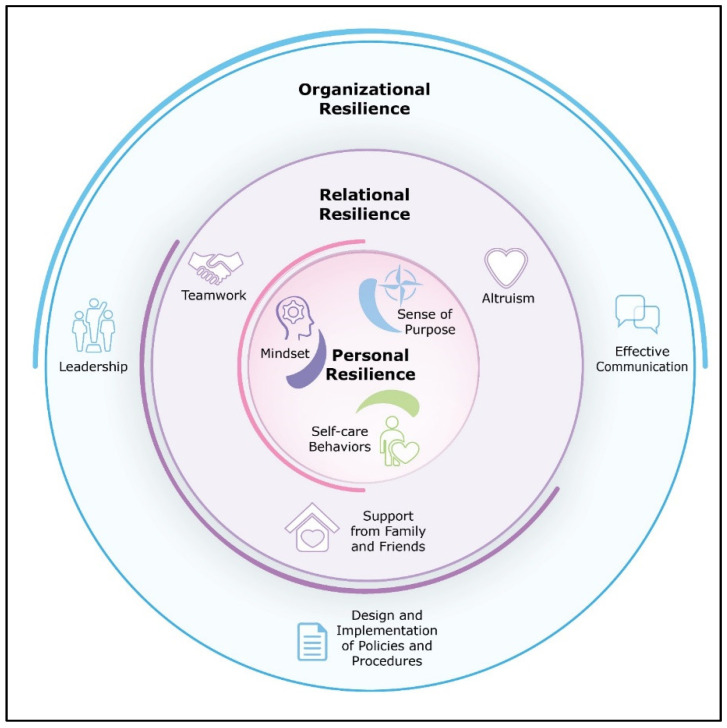
Sources of frontline healthcare worker resilience during COVID-19.

**Table 1 healthcare-09-01699-t001:** Participant demographic characteristics.

Variable	*N* (%)
Gender	
Female	16 (80%)
Age range	
30–39	9 (45%)
40–49	8 (40%)
50–59	3 (15%)
Profession	
Nurse	8 (40%)
Allied health	7 (35%)
Doctor	5 (25%)
Hospital setting *	
Acute care	80%
Rehabilitation	20%
Aged care	20%
Years in profession	
0–5	2 (10%)
6–10	1 (5%)
10–15	7 (35%)
16+	10 (50%)
Employment status	
Full-time	14 (70%)
Part-time or casual	6 (30%)
PSS	M = 12.7 (SD = 6.2)
Resilience	M = 32.1 (SD = 5.3)

* note that participants could work in more than one hospital setting; PSS: Perceived Stress Scale); CDRS: Connor–Davidson Resilience Scale (range 0–40).

**Table 2 healthcare-09-01699-t002:** Strategies to promote resilience.

**Individual Level**
Help healthcare professionals to cultivate a positive mindset (targeted sources of resilience: positive mindset and teamwork) Develop resources that help healthcare professionals challenge unhelpful thinking patterns and foster flexible thinking and realistic hope (e.g., by using expert-led resilience training sessions; provision of handouts; psychoeducation). Provide opportunities for debriefing to help frontline healthcare professionals process and make sense of their experiences (this arguably counterbalances an ‘action-oriented approach’ used when frontline healthcare professionals report having little time to think).
Provide opportunities to reflect on the value and meaning of pandemic work (targeted sources of resilience: finding purpose, altruism, positive mindset, and teamwork) Create forums to explore the meaning of pandemic work (e.g., invite brief comments on meaningful moments at clinical handover meetings; capture healthcare professional stories and share on social media and in organisation communications; post messages of gratitude in staff areas and websites) Share stories of frontline healthcare professional kindness (e.g., invite staff to write inspiring stories on staffroom notice board, share stories on organisation websites and via email)
Support healthy lifestyle choices (targeted sources of resilience: healthy lifestyle and teamwork) Promote physical activity at work (e.g., lunchtime walking, tai chi, Pilates, yoga) Support contemplative practices at work (e.g., staff mindfulness training, yoga or tai chi drop-in sessions) Provide access to well-being resources that can be used outside of work (e.g., subscription to mindfulness or well-being smartphone apps, subsidised gym memberships, personal trainers)
**Relational Level**
Recognise frontline healthcare professional external social supports (targeted source of resilience: social support) Provide adequate provisions for families (e.g., childcare support, caregiver leave, psychological assistance programs, add specific examples) Acknowledge healthcare professional social networks (e.g., invitations to organisation-led events, add specific examples)
**Organisational Level**
Develop effective policies and procedures (targeted source of resilience: design and implementation of policies and procedures, leadership, and effective organisation-wide communication) Establish a clear and logical chain of command regarding COVID-19 decision making, such as a ‘COVID command centre’ [16], and designated leaders within the organisation where responsibilities for policy development are clearly defined Prioritise strategies to instill healthcare professional trust in policies and procedures (e.g., timely provision of rationale for any changes in policy, forums for discussion and feedback, and engagement in priority setting at all stages)
Enhance leadership (targeted source of resilience: leadership and effective communication) Support ‘leading from the front’ (e.g., help leaders stay connected to their teams whilst acknowledging workload and infection control constraints during a pandemic; consider benefits of having physical or virtual leadership presence in COVID wards) Provide training opportunities for leadership development, with a focus on skills to care for staff emotional needs Provide support for leaders to decompress and manage their own mental health
Invest in hospital-wide communication platforms (targeted source of resilience: effective communication, implementation of policies and procedures, teamwork, and inspiring leadership) Co-ordinate multiple communication platforms for rapid dissemination of information to support the concept that “We are all in this together” (e.g., coordinating email, Zoom, face to face, newsletter e-bulletins, and whiteboard or printed flyers in clinical areas) Consider social communication platforms to help with rapid dissemination of information, whilst facilitating team bonding (e.g., WhatsApp, Facebook)

## Data Availability

Meta-data will be available for this study on the La Trobe University Figshare public access site. Individual transcripts cannot be released as they are potentially identifiable.

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
