# Peer review of "Sources of Resilience in Frontline Health Professionals during COVID-19"

_healthcare, 2021, doi:10.3390/healthcare9121699_

Round 1

Reviewer 1 Report

This study qualitatively investigates the resilience factors of HCWs struggling with COVID-19.

  1. In the introduction, the authors report some comments and some studies on HCWs. As is known, the mental health of HCWs struggling with COVID-19 has stimulated the interest of researchers, and, so far, more than 2000 articles and about 200 systematic reviews and meta-analyses have been published on this topic. The authors certainly cannot summarize all these studies, but they only report research that refers to the first phase of the pandemic, when the lack of personal protective equipment, lack of confidence in safety measures and fear of infections were predominant. In nearly two years, the picture of psychosocial factors has radically changed. Gradually isolation, compassion fatigue, monotony became important. The authors should more correctly indicate this evolution of the problems, which generate different resilience techniques corresponding to it. It is advisable to evaluate the results of the first longitudinal study that followed the frontline HCWs for one year: Magnavita, N.; Soave, P.M.; Antonelli, M. A One-Year Prospective Study of Work-Related Mental Health in the Intensivists of a COVID-19 Hub Hospital. Int. J. Environ. Res. Public Health 2021, 18, 9888. https://doi.org/10.3390/ijerph18189888.
  2. A comparison can also be made with workers interviewed in the same period in which they conducted the research, on the end of 2020: Magnavita N, Soave PM, Antonelli M. Prolonged Stress Causes Depression in Frontline Workers Facing the COVID-19 Pandemic-A Repeated Cross-Sectional Study in a COVID-19 Hub-Hospital in Central Italy. Int J Environ Res Public Health. 2021 Jul 8;18(14):7316. doi: 10.3390/ijerph18147316.
  3. A general problem of the study is generated by the definition of frontline workers. How did the authors classify HCWs as "frontline"? I have strong doubts that physiotherapists, occupational therapists and psychologists have dealt with various COVID-19 patients. Since a quarter of respondents have never dealt with SARS-CoV-2 infected patients, and the others have seen some in the past, but none of them is working in a COVID-19 hospital, they cannot be defined as "frontline". I would recommend removing this term from the title. The study is interesting even if it refers to common health workers.
  4. The choice of a convenience sample, 20 workers from 5 hospitals, who did not deal with COVID-19 patients or who chose not to work in the COVID-19 ward is a strong limitation that needs to be reported.
  5. The advice for increasing resilience that is reported in the Discussion is appropriate when referring to a heterogeneous group of healthcare professionals who do not have daily relationships with COVID-19 patients. The work in a COVID-19 hub-hospital such as the one described in the Italian longitudinal study cited above takes place in conditions of work overload and lack of time for physical activity, meditation and prayer. The authors should clarify that their study refers to generic (non-frontline) healthcare workers and that the advice provided hardly applies to situations where workers are overworked and do not have time for themselves.

Reviewer 2 Report

Dear colleagues, I hope this message find you well.

Thank you for giving me the opportunity of reading the work “Sources of Resilience in Frontline Health Professionals During COVID-19, it has been a very big pleasure to collaborate reviewing this manuscript. The topic of this paper is very interesting and it seems necessary to delve it. However, there are several questions to improve before to publish it. I would suggest some changes: 

Introduction

  • The structure of the introduction is not clear. I recommended to divide the introduction into several subsections. For example, creating a specific subsection where describe each variable involved.
  • On the other hand, when you explain the signs of psychological distress and mental health as a result of COVID-19, it is necessary to add more data. I recommend you to add this paper recently published: https://doi.org/10.3390/ijerph18147422
  • Moreover, introducing more studies could be interesting in order to support better how COVID-19 has psychologically affected health-care professionals:

https://doi.org/10.3390/jcm10184077

https://doi.org/10.1016/S2215-0366(20)30307-2

  • Aims and hypotheses should be detailed specifically.

Method

  •  

Results

  • Ok

Discussion

  • In my humble opinion, it could be useful to describe in more detail the practical and theoretical implications of this research. It would be useful they contextualize better the contribution within the framework of the issue explaining why the contribution is useful and enrich the impact.
  • Limitations should be explained more deeply.

Conclusions

  • Ok

Round 2

Reviewer 1 Report

The authors have improved the work by accepting the suggestions proposed. However, they did not submit the full version of references and did not put the references in order

This manuscript is a resubmission of an earlier submission. The following is a list of the peer review reports and author responses from that submission.